# Quantum simulations made easy plane

Andreas M. Läuchli

*Institute for Theoretical Physics, University of Innsbruck, A-6020 Innsbruck, Austria*

R. Moessner

*Max Planck Institut für Physik komplexer Systeme, D-01187 Dresden, Germany*

(Dated: July 16, 2018)

Ever since Heisenberg's proposal of a quantum-mechanical origin of ferromagnetism in 1928, the spin model named after him has been central to advances in magnetism, featuring in proposals of novel many-body states such as antiferromagnets, emergent gauge fields in their confined (valence bond crystal) and deconfined (resonating valence bond spin liquids) versions. Between them, these cover much of our understanding of modern magnetism specifically and topological states of matter in general. Many exciting phenomena predicted theoretically still await experimental realisation, and cold atomic systems hold the promise of acting as analogue 'quantum simulators' of the relevant theoretical models, for which ingenious and intricate set-ups have been proposed. Here, we identify a new class of *particularly simple quantum simulators* exhibiting many such phenomena but obviating the need for fine-tuning and for amplifying perturbatively weak superexchange or longer-range interactions. Instead they require only moderate *on-site* interactions on top of uncorrelated, *one-body* hopping–ingredients already available with present experimental technology. Between them, they realise some of the most interesting phenomena, such as emergent synthetic gauge fields, resonating valence bond phases, and even the celebrated yet enigmatic spin liquid phase of the kagome lattice.

Quantum spin liquids and other exotic magnetic states are among the theoretically most interesting yet experimentally elusive instance of collective quantum phenomena, underpinning as they do the counterintuitive emergence of new particles from the 'decay' of electrons in a magnet [1], such as holons and spinons which carry the electric and magnetic properties of an electron independently, or the entirely novel Majorana particles underpinning attemps to build so-called topological quantum computers protected against decoherence due to noise from the environment [2].

Some of the earliest proposals for quantum simulators have thus aimed at generating such exotic spin physics, e.g. in the case of the Kitaev model [3] with its topological spin liquid phase [4]. The challenges facing such a programme are manifold and formidable. Firstly, it is necessary to generate the requisite degrees of freedom; secondly, they need to inhabit a suitable optical lattice; thirdly, their interactions must be non-trivial and satisfy symmetries of the target Hamiltonian; and finally, the system in question must be cooled to a temperature below which the target many-body states are stable. All these points have seen much progress on the theoretical front over the years, and considerable effort has been invested in identifying promising settings in which naturally occurring strong interactions – be it e.g. of dipolar nature or through Rydberg physics – can lead to interesting many-body states.

Here we identify a class of particularly simple spin one-half model Hamiltonians which go along with particularly – we believe, maximally – simple experimental settings, for which the first three challenges are all comfortably in the realm of current technology. Our striking underlying technical observation is that some of the most interesting collective phenomena predicted for $S=1/2$ Heisenberg (HB) spin models are in fact not predicated on the SU(2) symmetry of Heisenberg models at all. Rather, they persist across a wide swathe of exchange anisotropy parameters, In cases of interest this includes XY models, which lend themselves to cold atomic settings, as the $S=1/2$ degree of freedom can simply be represented by the presence or absence of a particle, requiring only moderate *on-site* interactions to forbid double or higher occupancy. This is much easier to generate than interactions between neighbours, which are only expected to become available once dipolar atoms or molecules get operational for quantum simulations. Due to the widespread experimental availability of the bosonic atoms $^{87}$Rb and $^{133}$Cs and the fact that in our proposal the spin exchange energy scale is equivalent to the strength of the bosonic hopping matrix element, we believe that the energy scales are very favourable for atomic quantum simulations.

The respective spin and bosonic Hamiltonians read

$$\mathcal{H}_{\text{spin}} = J \sum_{\langle i,j \rangle} (S_i^+ S_j^- + S_i^- S_j^+)/2 + \Delta\, S_i^z S_j^z \qquad (1)$$

$$\mathcal{H}_{\text{boson}} = J \sum_{\langle i,j \rangle} (b_i^\dagger b_j + b_i b_j^\dagger) + \frac{U}{2} \sum_i n_i(n_i - 1)\,,$$

where $S_j^z$ and $S_j^\pm = S_j^x \pm i S_j^y$ are spin-1/2 operators and $b_j^\dagger$ and $n_j = b_j^\dagger b_j$ are boson operators at site $j$ of a lattice defined by its bonds $\langle ij \rangle$. We consider antiferromagnetic spin-spin interactions $J > 0$ in the following.

The anisotropy parameter $\Delta = 1$ for the HB model. Crucially, it vanishes for the XY case, corresponding to $\mathcal{H}_{\text{boson}}$ in the limit of large $U$. It can thus be realised containing a simple hopping term – that is to say a *one-body* kinetic term! Thus, the problem is reduced to one of engineering an optical lattice for the hardcore bosons – i.e. with only *onsite* interactions – with $J > 0$!

The mapping between hardcore bosons and $S=1/2$ XY models is well known and the intrinsic interest of hardcore bosonic models with the hopping $J$ as the magnetic energy scale has also been recognised [5–10]. Our main contribution

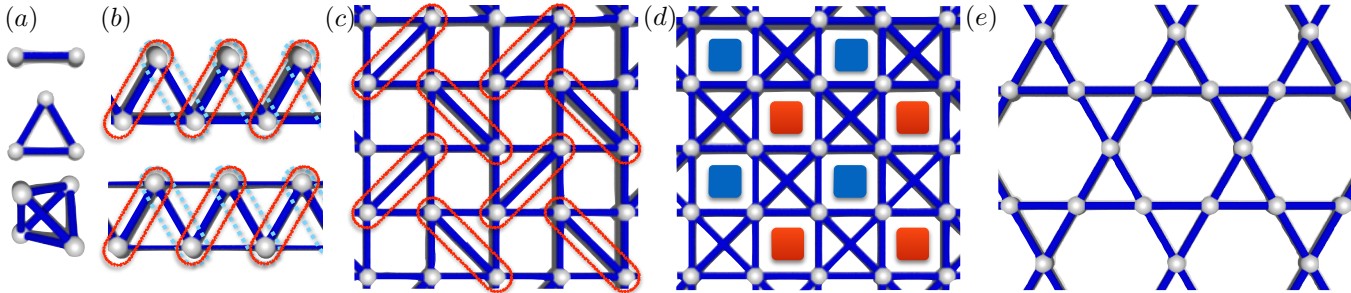

FIG. 1. (Color online) Equivalence between ground states of HB and XY models. This holds exactly for (a) connected clusters such as dimers, triangles and tetrahedra, as well as (b) the sawtooth and Majumdar-Ghosh chains, and (c) the Shastry-Sutherland lattice, in which spins form singlet pairs (denoted in red). It is approximate but accurate for the archetypal frustrated models on the well-studied (d) checkerboard (cf. Fig. 2) and (e) kagome lattices, including the emergent synthetic gauge field of the former, and the celebrated spin liquid state of the latter. No simple picture for the ground state of the kagome magnet is known.

is to explicitly demonstrate that a number of very interesting phenomena do in fact occur in such XY models, including the physics of an emergent synthetic gauge field, the realisation of resonating valence bond physics and the appearance of quantum spin liquidity, topics which have been at the focus of intense attention recently.

We first present an origin of this remarkable XY-HB 'equivalence', and present families of frustrated lattices where it applies, identifying the kagome quantum spin liquid as perhaps the most exciting and promising target for realisation via quantum simulators. We complement this by addressing experimental realisability with current AMO technology, and conclude with an outlook.

*Origin of XY-HB equivalence:* Our basic idea can already crisply be gleaned from considering a fully connected cluster of $q$ spins, where $q = 3(4)$ corresponds to a triangle (tetrahedron). Up to a constant, the Hamiltonian $\mathcal{H}_{\text{spin}}$ for the cluster can be written in terms of its total spin, $\mathbf{L} = \sum_{i=1}^{q} \mathbf{S}_i$:

$$H_{HB} = \mathbf{L}^2 = L(L+1) \tag{2}$$
$$H_{XY} = \mathbf{L}^2 - L_z^2 = L(L+1) - L_z^2 \tag{3}$$

with $L$ a (half-)integer between $L_{min} = 0$ $(1/2)$ and $L_{max} = q/2$ for $q$ even (odd), and $-L \leq L_z \leq L$ in integer steps.

Thus, *for both HB and XY spins* the ground state always has $L = L_z = L_{min}$. In both cases – indeed, for any value of $\Delta$ – all states can be completely labelled by assigning values $\{L, L_z\}$ but their order in energy above the ground state differs between HB and XY – the equivalence only holds at low energy. We believe this is an important ingredient for the quasi-equivalence of low-energy states of XY and HB more generally. We next demonstrate the HB-XY equivalence explicitly, for a number of different lattices (Fig. 1), among them some of the most studied exotic magnets.

*Sawtooth and Majumdar-Ghosh chains:* The first instances are two triangle-based chains, Fig. 1(b), where the HB and XY groundstates are exactly identical. The ground states are exact valence bond coverings for the HB case, in which neighbouring spins are paired up into SU(2) singlets, which are also XY ground states.

*Shastry-Sutherland Lattice:* The next exhibit is the Shastry-Sutherland [11] lattice, Fig. 1(c), which has received much attention in the context of the material $SrCu_2(BO_3)_2$ [12]. Two particular highlights of the Shastry-Sutherland lattice are i) its exact valence bond covering ground state for sufficiently strong diagonal coupling (going back to an exact solution by Shastry and Sutherland [11]) and ii) a fascinatingly rich magnetization plateaux structure [13] exhibiting e.g. crystals of triplon bound states and even spin supersolids [14].

We performed numerical exact diagonalizations of finite XY systems and find that the exact singlet covering ground state exists when the coupling on the diagonal bonds exceeds $\sim 1.6$ times the coupling on the square lattice bonds. For smaller diagonal couplings a Néel state in the spin $x-y$ plane is formed. This state is equivalent to a staggered superfluid state of the hardcore bosons. Both phases are also present in analogous form in the HB phase diagram at zero magnetization [15, 16].

*Checkerboard Lattice:* The next case, whose analysis is considerably less straightforward, is the checkerboard lattice [Fig. 1(d)] also known as the two-dimensional version of the pyrochlore lattice. Both can be thought of consisting of tetrahedra arranged to share corners, with a tetrahedron projected onto a plane corresponding to a square with diagonal interactions [Fig. 1(a), bottom]. The Heisenberg model on the checkerboard lattice has been established to exhibit quantum order by disorder, in the form of a plaquette-type valence bond crystal [17–20].

We have confirmed from finite-size numerical diagonalization studies that for both XY and HB cases, the same ground-state correlations – in particular, the same symmetry-broken non-classical plaquette valence-bond crystal – occurs, see Fig. 2.

*Kagome Lattice:* Perhaps the most surprising case of our HB-XY equivalence is that of the kagome lattice, Fig. 1(e). The $S=1/2$ HB antiferromagnet on this lattice has become one of the paradigms of the physics of quantum spin liquids. Despite intense theoretical efforts for more than twenty

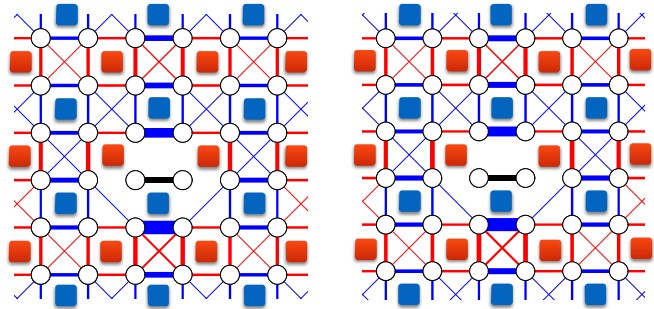

FIG. 2. (Color online) Valence bond crystal on the checkerboard lattice. The two panels show the essentially identical pattern ground state correlations for $N = 36$ spins for the XY (left) and HB (right) models. The thickness of the red(blue) lines indicates the degree of suppression(enhancement) of the probability of finding a valence bond given the presence of the black valence bond, showing the enhancement on the plaquettes as in Fig 1(d).

years [21–27], further reinforced by the suggestion that with Herbertsmithite there may or may not be a material available which possibly realizes a HB $S=1/2$ model with only small deviations [28–30], the nature of the ground state and the low lying excitations is still not settled. Thus, unlike the previous examples, no reliable solution of the kagome $S=1/2$ HB model is known, either exact or approximate, and it is here that results from quantum simulations would be most welcome.

The XY-HB equivalence is most starkly illustrated by considering the nature of the low-energy states of the two models for a finite-size system. Compared to the case of the clusters discussed above, where the quantum numbers $\{L, L_z\}$ were used to establish a correspondence between the states for XY and HB, for a lattice system, the quantum numbers are richer. The states can be grouped into sectors, labelled by the irreducible representations of the space group of the lattice, with each sector now containing an exponentially large number of states. The spectrum thus decomposed is shown in Fig. 3 for a cluster of 36 sites, where we have effected a simple overall rescaling of the energy axis, multiplying the excitation energies for XY compared to HB by a factor of 2.

We thus find that there is a precise pairwise XY-HB correspondence between not only the ground [31] but also *each and every* low-energy excited states in *each and every* sector! *The entire low-energy spectrum is in near-perfect correspondence between HB and XY cases!!!*

*Quantum Simulators:* With this in mind, we now turn to prospects for realising these models in actual cold atomic quantum simulators, focussing our attention on the cases for which the exact ground states are not analytically available. In particular, given the combination of the great interest in the kagome spin liquid phase and its faithful representation by an XY model, we believe that this establishes the hardcore boson model on the kagome lattice alongside the checkerboard lattice with its emergent gauge field as the most promising target quantum simulator for exotic spin physics. As mentioned above, all we need is (a) band structure engineering and (b)

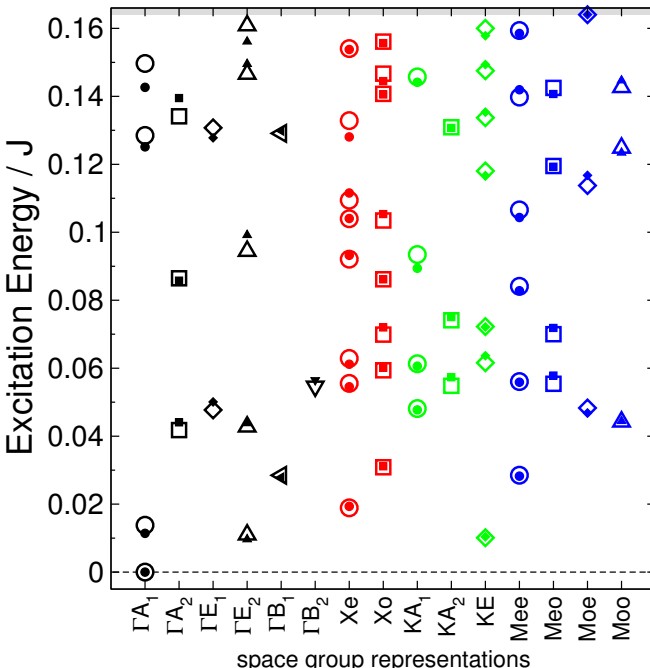

FIG. 3. (Color online) Kagome lattice: The low-lying many body energy spectrum on $N = 36$ site sample for HB (open symbols) and XY (solid symbols) is identical modulo a single global shift and rescaling.

hardcore repulsion.

Regarding (a), optical kagome lattices have already been obtained and studied [32]. The sign of the hopping, which needs to be 'frustrated' for the spin liquid to appear instead of a conventional superfluid can be obtained by the by now well-understood protocol involving shaking [33–36], therefore changing the signs of all three hopping integrals at once. Another promising setup is to alter the sign of the hopping along only one of the three linear chain directions of the kagome lattice through Raman-assisted hopping [37], as recently experimentally demonstrated in ladder [38] and square lattice Hofstadter systems [39, 40]. This amounts to a different gauge choice to implement the physical flux $\pi$ (equivalent to the frustrated hopping) through each triangle. For the checkerboard lattice, we are not aware of any experimental attempts using cold atoms so far, although promising proposals in this direction making use of the properties of Rydberg atoms have been put forward [41]. Our proposal supplements these by identifying an avenue requiring no fine-tuning of either interactions or hopping – rather, it suffices to generate a square optical lattice, decorated with the additional checkerboard diagonal hoppings of strength in the range $0.75 \sim 1.25$, according to the HB results [42], without need for particular fine-tuning.

Regarding (b), we note that it is not even necessary to reach the ideal limit of hardcore bosons to access the phases of the XY model. This is true for both checkerboard valence bond crystal [41] and the kagome spin liquid state, for which

(Fig. 4) the overlap of the exact XY ground state with that of a model with softened onsite repulsion, $U$, is high for $U/J > 10$, and almost perfect for $U/J > 20$. These conditions are easily met with either $^{87}$Rb or $^{133}$Cs atoms. $^{133}$Cs has the further advantage of a Feshbach resonance, allowing to access a large $U/J$ ratio while maintaining a sizeable hopping energy scale $J$.

With our proposed kagome quantum simulator one can then first address interesting few particle physics, such as the dynamics of caged (localized) magnons [43] which could be monitored by single site resolution, then proceed to the investigation of spontaneous pattern formation in various magnetisation plateaux [44, 45], which could be detected using Bragg spectroscopy, and then ultimately address the nature of the spin liquid at half filling, where many theoretical questions are still open.

How does our proposal help in terms of 'practicalities' of reaching a regime where the cooperative physics of these models is visible? The challenges here are very similar for most proposals of atomic quantum simulators, with our proposal here perhaps favourable in the following ways. Firstly, we invoke exclusively on-site – and therefore generically strong – interaction terms rather than perturbatively suppressed 'second-order' superexchange terms, or weak further-neighbour or fine-tuned interactions. Secondly, the other energy scale is given by *one-body* hopping terms, which can be implemented as 'first-order' terms for the kagome lattice, and which should therefore be comparatively large. Thirdly, since the ratio of hardcore repulsion to hopping does not need to be all that large, the different terms in the Hamiltonian do not need span an exorbitant range of energies by themselves. Taken together, these provide a high level of robustness against coupling to the environment or experimental noise, which will e.g. allow a larger time-window of stability of the system in order to study its real-time dynamics. Regarding cooling, while for a strict fixed-entropy preparation protocol, a larger energy scale makes no difference, it can be very useful when using a spatially inhomogeneous setting in which another part of the system acts as a 'bath' for the quantum simulator. At any rate, a study of the finite-temperature properties of the kagome spin liquid is already be a worthy goal in itself.

*Summary and outlook:* In summary, we have proposed a set of maximally simple quantum simulators to capture the interesting many-body physics of frustrated $S=1/2$ HB magnets. This include various poster children of correlated electron physics and topological phases of condensed matter in the form of resonating valence bond phases and all the way up to the enigmatic kagome spin liquid. The observation of the persistent plaquette valence-bond crystal in the XY case is all the more intriguing when considering that this phase is adiabatically connected to the strong Ising limit, where the XXZ Hamiltonian maps onto quantum square ice, a quantum link model realising the $U(1)$ lattice gauge theory (LGT) which has an emergent gauge field, albeit confined at zero temperature in $2D$. It should thus be feasible to study LGTs and

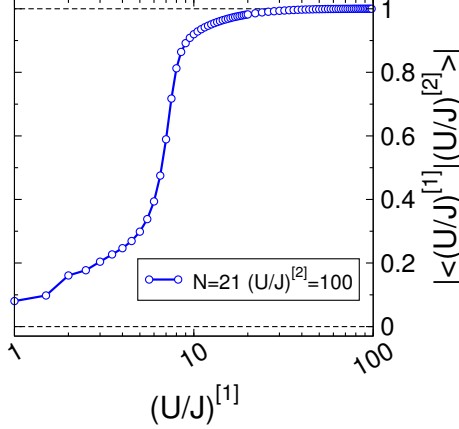

FIG. 4. (Color online) Kagome lattice: Overlap of the ground state at $U/J = 100$ with the ground state at lower $U/J$. Down to values of $U/J \sim 10 - 20$ the spin liquid physics will prevail over competing phases.

their emergent gauge fields through implementations that are much simpler than those which attempt to enforce the gauge constraint in a strict way.

Many further avenues for research are plainly visible. To start with, the robustness of the kagome spin liquid phase, which persists for a fabulously broad range of transverse couplings, obviously requires a deeper understanding. More broadly, the quantum simulation of XY magnetisation curves – by tuning the density of bosons – would link up with the rich and as yet not fully understood field (again, especially for kagome magnets [44, 45]) of magnetisation plateau in frustrated quantum magnets.

Moreover, we also expect an XY-HB analogy for a wide range of other lattices which have received considerable interest over the years in $d = 1, 2$ and 3, such as the kagome strip [46], the square kagome (squagome) lattice [47] and even the hyperkagome lattice [48]. For the Husimi cactus [49], which has played an important role for understanding excitations in kagome-type lattice, but is hard to represent as an optical lattice, there is an exact equivalence between *exponentially many* XY and HB ground states.

More conceptually, note that perturbative schemes based on expansions around singlet coverings of lattices like the one used to derive the Rokhsar-Kivelson Hamiltonian [50] of short-range resonating valence bond physics [51] will apply not only for HB but also to their corresponding XY models. On general grounds, one then expects that *gapped* phases, like a topological $Z_2$ spin liquid [52] with SU(2) symmetry [53], will then be stable as one perturbs from the HB towards the XY point.

Beyond this, the potential applicability of these ideas for the interesting cases of pyrochlore (the $d = 3$ brethren of the checkerboard) or the SCGO lattice (a hybrid between pyrochlore and kagome) is entirely open. For these, there are at present *no* reliable analytic or numerical approaches available, and therefore, cold atomic quantum simulators could estab-

lish themselves as the method of choice for studying complex higher-dimensional quantum magnets.

Overall, exploring the physics of frustrated $S=1/2$ XY models via hardcore bosons on optical lattices patently has tremendous potential for advancing our understanding of, and experimental access to, the physics of quantum spin liquids, emergent synthetic gauge fields, and other exotic phenomena in magnetic quantum matter.

## ACKNOWLEDGEMENTS

We thank N.R. Cooper, C. Hooley and S.L. Sondhi for useful comments and I. Bloch, F. Meinert and H.-C. Nägerl for discussion. AML was supported by the FWF (I-1310-N27/DFG FOR1807) and FWF SFB Focus (F-4018-N23) and HPC resources at MPG RZ Garching, UIBK and VSC2/3.

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
