# Peer review of "Quantum simulations made easy plane"

_SciPost Physics_

## Round 1 · Referee Report · Anonymous (Referee 1) · 2016-12-30

Strengths

1.- New setup for quantum the quantum simulation of interesting phases of matter, many of them hard to assess with current numerical methods.

2.- The setup is in principle possible with current technology

Weaknesses

1.- A more detailed explanation of the experimental details for the proposed optical-lattice setup would be desirable.

2.- More details on the numerical simulations supporting the main claims of the paper (namely, the mapping between Heisenberg and XY eigenstates) would also be desirable.

Report

In this paper the authors point out that many interesting regimes of the Heisenberg model in a variety of lattices can be reproduced by an XY model which, in turn, can be simulated with current technology by hopping bosons in optical lattices. This is a rather simple but somehow nontrivial and, because of some reason, overlooked observation. The authors support theor claims with several numerical simulations computing the spectrum of the Heisenberg and XY models in a variety of setups, showing the mapping between the low-energy sectors of these models. I think this is an interesting observation that has the potential to be useful in a near future, in the context of optical-lattce quantum simulators. Therefore I recommend the paper for publication in SciPost.

Requested changes

Before publication, I would like the authors to show more numerical results further supporting their claims, via exact diagonalization and/or other methods, as well as some more details on the proposed optical lattice setup. In this sense, perhaps a longer version of the paper, with a more detailed discussion, would be desirable.

---

## Editorial Decision

awaiting_resubmission